# Mesenchymal and Induced Pluripotent Stem Cells-Derived Extracellular Vesicles: The New Frontier for Regenerative Medicine?

**DOI:** 10.3390/cells9051163

**Published:** 2020-05-08

**Authors:** Maria Magdalena Barreca, Patrizia Cancemi, Fabiana Geraci

**Affiliations:** 1Section of Biology and Genetics, Department of Biomedicine, Neuroscience and Advanced Diagnostics (Bi.N.D.), University of Palermo, Via Divisi 83, 90133 Palermo, Italy; mariamagdalena.barreca@unipa.it; 2Department of Biological, Chemical and Pharmaceutical Sciences and Technologies (STEBICEF), University of Palermo, Viale delle Scienze Bld. 16, 90128 Palermo, Italy; patrizia.cancemi@unipa.it

**Keywords:** extracellular vesicles, stem cells, mesenchymal stem cells (MSCs), induced pluripotent stem cells (iPSCs), regenerative medicine

## Abstract

Regenerative medicine aims to repair damaged, tissues or organs for the treatment of various diseases, which have been poorly managed with conventional drugs and medical procedures. To date, multimodal regenerative methods include transplant of healthy organs, tissues, or cells, body stimulation to activate a self-healing response in damaged tissues, as well as the combined use of cells and bio-degradable scaffold to obtain functional tissues. Certainly, stem cells are promising tools in regenerative medicine due to their ability to induce de novo tissue formation and/or promote organ repair and regeneration. Currently, several studies have shown that the beneficial stem cell effects, especially for mesenchymal stem cells (MSCs) and induced pluripotent stem cells (iPSCs) in damaged tissue restore are not dependent on their engraftment and differentiation on the injury site, but rather to their paracrine activity. It is now well known that paracrine action of stem cells is due to their ability to release extracellular vesicles (EVs). EVs play a fundamental role in cell-to-cell communication and are directly involved in tissue regeneration. In the present review, we tried to summarize the molecular mechanisms through which MSCs and iPSCs-derived EVs carry out their therapeutic action and their possible application for the treatment of several diseases.

## 1. Introduction

Differently from lower vertebrates, such as zebrafish and amphibians, humans have a limited ability to regenerate damaged tissues or organs, restoring their original state. To date, the clinical strategies to recover organ or tissue function fall into three main categories: drug therapy, auto or heterotransplants, and cell therapy and tissue engineering. More than 15 years ago, the term “regenerative medicine” entered into our scientific lexicon. It is a new interdisciplinary branch of medicine that develops methods to regrow, repair or replace cells, organs or tissues damaged by age, disease, or trauma, as well as to normalize congenital defects.

Precursors of regenerative medicine can be considered cell or organ transplants. More than half a century ago the first successful organ transplantation was performed in Boston [1], and it has been the cornerstone therapy for replacing diseased or malfunctioning ones. Besides, in the same period, the first bone marrow transplant was performed [2]. The principal downsides with organ transplantation includes the lack of donors, the immunological compatibility and the immune suppression to avoid organ rejection.

A great stimulus to regenerative medicine derives from the discovery of stem cells more than four decades ago, thanks to their ability to self-renew and differentiate into a variety of cell lineages. In fact, the two main components of regenerative medicine are stem cell-based therapy, either endogenous or injected, and tissue engineering regenerative medicine, based on the use of biomaterials alone or seeded with stem cells. Regenerative medicine minimalizes the problem of transplanted organ rejection. In several human diseases, stem cells have been successfully applied, especially in the hematological field [3,4,5], even if stem cell therapy has not yet reached the level of solid organ regeneration.

## 2. Stem Cells

Stem cells are undifferentiated cells characterized by their self-renewal capability, the ability to divide generating cells equal to themselves, and by their competence to give rise to specialized cells. According to their differentiation potential they could be classified into: totipotent stem cells, which are able to differentiate in all the body cell types plus the extra-embryonic cells [6]; pluripotent cells that are able to give rise to all of cell types of the body [7,8]; multipotent stem cells, which can develop several cell types within one particular lineage [9], and finally, unipotent cells responsible for the differentiation of only one cell type [10,11]. Moreover, depending on their origin, stem cells are classified into embryonic (ESCs), fetal (e.g., umbilical stem cells, amniotic stem cells), adult, and induced pluripotent stem cells (iPSCs) [12,13]. The latter are pluripotent stem cells derived from adult somatic cells, genetically reprogrammed to an embryonic stem cell-like state [14], with self-renewal and differentiation capability, but free from the ethical issues that afflicted ESC use. The first successful reprogramming of human somatic cells was reported in 2007 by Takahashi and co-workers [15]. Initially, the reprogramming of somatic cells was obtained retrovirally introducing four key transcription factors that are responsible for pluripotency maintaining (i.e., Oct3/4, Sox2, Klf4, and c-Myc). Nowadays, different and less dangerous methods to introduce the “Yamanaka factors” within somatic cells have been successful applied [16,17,18,19].

## 3. Stem Cells and Regenerative Medicine

Over the past 20 years, much attention has been paid to stem cell biology, resulting in an extensive understanding of their characteristics and therapeutic application potential [20].

The application of stem cells in regenerative medicine mainly regards hematological disorders and skin regeneration [21,22]. Already in 1984, Gallico et al. demonstrated that the human epidermal cells isolated from a skin biopsy were able to perform epithelial sheets, which when implanted on burn wounds generated a permanent epidermidis [21]. Today it is well known that the epidermidis undergoes continuous renewal due to the presence of a special population of keratinocytes stem cells, which could be massively expanded in culture [23]. To date, substantial progresses have been made in human keratinocytes culture methods prior to be implanted on extensive third degree burn wound. In particular, Ronfard et al. demonstrated that fibrin matrices make epithelia very easy to handle. Moreover, they maintain their original dimensions when detached, and they are also three times larger than grafts obtained without fibrin matrix [24]. Autologous transgenic keratinocytes were also used to regenerate the entire epidermidis in a patient suffering from junctional epidermolysis bullosa, a genetic defect causing chronic wounds in both skin and mucosa [25].

Today, several multipotent stem cells of different sources and iPSCs have been introduced for tissue repair. Some preclinical studies and clinical trials confirmed the potential of stem cell therapy in treating several diseases, such as diabetes mellitus [26,27], heart failure [28,29], disorders of the nervous system [30,31,32], and so on. Particularly attracting as candidates for regenerative medicine are mesenchymal stromal stem cell (MSCs) for their anti-inflammatory and immunomodulatory features [33,34,35]. MSCs do not express significant histocompatibility complexes and immune stimulating molecules, are not detected by immune surveillance, and do not lead to graft rejection after transplantation. In addition, MSCs display a great homing potential and migration capability to the damaged sites [36]. Their potential in regenerative medicine is also due their ability to enhance angiogenesis and accelerate tissue healing [37]. MSCs can easily undergo multilineage differentiation, such as adipocytes, hepatocytes, osteoblasts, chondrocytes, myoblasts, and neuronal cells. [38]. They can be isolated from almost all adult tissues, such as bone marrow, adipose tissue, synovium, dental pulp, from perinatal tissues, such as umbilical cord blood, placenta, amniotic fluid, and umbilical cord Wharton’s jelly [39,40,41,42,43,44,45,46], as well as from solid organs, e.g., brain, lung, liver, pancreas. Adipose derived mesenchymal stem cells (ADSCs) show several advantages over other MSCs. ADSCs are easily accessible and their extraction is technically easy and safe, with a higher proliferative capacity [47,48,49]. All these advantages make ADSCs optimal for stem cell-based therapies [50,51,52].

Stem cells (i.e., bone marrow derived stem cells) were successfully used for improving wound repair of skin radiation burns, a common radiotherapy complication. Conventional surgical treatment coupled with stem cell therapy (i.e., autologous MSCs). was for the first time applied on a human patient by Lataillade et al. [53]. Following this combined therapy, they observed that the healing of the lesion proceeded easily and that there were no side effects. Moreover, the healing persisted one year after the procedure. The researchers hypothesized that MSC effects on the radiation burn healing could be mediated by the secretion of soluble mediators, cytokines, and trophic factors that may play important roles to counteract the local inflammatory processes [53]. A similar positive result was obtained two years later by Bey et al. [54].

The positive effects of MSCs in severe radiation burns were also demonstrated by Linard et al., as they observed the positive outcomes of stem cell utilization in preclinical pig model of radiation induced proctitis and of severe radiation burn [55,56]. In the former model, MSC injection markedly reduced tissue inflammation by diminishing infiltrates of macrophages and changing their phenotype towards M2 anti-inflammatory macrophages [56]. 

After severe irradiation, exposure lesions may extend beyond the skin affecting the underlying muscles as well. In this case, MSC therapy stimulates the angiogenic factor VEGF, which is involved in muscle cell regeneration and growth, and a macrophage M1/M2 balance [55].

Differently from bone marrow and cord blood transplantation in leukemia or other hematological disorder treatment, the use of other stem cells, including keratinocytes used for skin regeneration, needs expansion. This process requires stem and progenitor cells to be preserved throughout all the culture steps. To this aim, several studies have been performed to identify the molecular stemness regulators. One of these regulators is the Klf4 gene, which is also one of the four factors responsible for somatic cell reprogramming to obtain IPSCs [15]. In skin therapy by keratinocytes the preservation of both stem and progenitor cells features is particularly important, as it is a necessary condition for successful long-term graft outcome. Fortunel et al. have demonstrated that Klf4 downregulation increased keratinocyte stem cells immaturity and its clonogenic potential, increasing their ex vivo expansion. In addition, Klf4 deficient cells displayed improved graft capacity in an in vivo xenograft model [57].

## 4. Stem Cells Drawbacks in Regenerative Medicine

Even if stem cell therapy displays great potential in the field of regenerative medicine and MSCs have been already successfully used in several clinical applications (see clinicaltrial.gov database) [58], their use also presents some risks, such as cell rejection, undesired immune response [59], possible contamination with viruses [60], low recovery rate [61,62,63], problematic transport and storage of cells before their use [64,65,66], and adverse effects associated with their harvest in the case of bone marrow mesenchymal stem cells. Other important risks of MSCs are tumorigenicity, proinflammation, and fibrosis. Moreover, for cell therapy applications, their ex vivo expansion is necessary, although a reduction in their effectiveness was observed during in vitro cultivation, due to aging, loss of stemness and undesired differentiation. Finally, it is well documented that proliferation and differentiation capability of MSCs are also negatively influenced by donor age. 

The discovery of iPSCs by Takahashi and collaborators [15] allows researchers to eliminate some of the disadvantage of MSCs. However, genetic instability along with plasticity and self-renewal, common to both MSCs and iPSCs, could lead to tumor formation in the host tissue [67,68,69,70]. Notwithstanding, regardless of the promise of stem cell research to treat untreatable disease, such as Parkinson’s disease, spinal cord injuries, macular degeneration, amyotrophic lateral sclerosis, their use as a routine therapy is still a long way off. Indeed, to exert their regenerative potential, stem cells need to home the injured site, engraft it, and due to their self-renewal and differentiation capability replace damaged cells. Unfortunately, numerous preclinical studies have shown a low grafting rate. Indeed, systemic stem cell delivery by intravenous injection determines the capture of the majority of cells into the capillary beds, especially in the lungs, but also in spleen, liver, and kidney. Because of the systemic clearance [58,71], only a small amount of stem cells reaches the target site [68,72,73,74,75,76,77]. These limitations could be overcome by directly injecting stem cells into the damaged site, e.g., direct myocardial infusion after infarction. Despite this, problems still remain for the successfully stem cell engraftment and in situ differentiation: for example a low cell survival rate was found in the injured site [78,79,80]. Furthermore, genetic instability of stem cells, especially iPSCs, along with plasticity and self-renewal could lead to tumor formation in the host tissue, should be still evaluated [67,68]. 

Experimental data obtained by animal models of myocardial infarction and brain stroke showed that ADSC treatment improved both cardiac and neurological function. However, there are no unambiguous evidences demonstrating that the amelioration of symptoms was due to ADSC transdifferentiation into functioning cardiomyocytes or neurons [78,81,82,83,84]. MSCs therapeutic effects might be due to the secretion of bioactive factors, collectively named secretome, as well as extracellular vesicles (EVs) [73,85,86,87,88,89,90,91,92,93,94,95,96]. The secretome consists of bioactive factors released by stem cells both actively and passively; it includes soluble protein factors (i.e., growth factors and cytokines) and EVs, which transfer to recipient cell proteins, lipids, and nucleic acid [97]. EV paracrine effects were firstly described by Gnecchi et al. [98]. MSC secretome composition is dependent on origin of the stem cells and culture conditions. Moreover, it is also influenced by preconditioning treatments of the stem cells during cell culture [95,99,100]. For example, Magne et al. demonstrated that IL-1β priming of gingival MSCs led to secretome changes, potentiating therapeutic effects in both in vivo and in vitro models of skin trauma. Their results showed an increase in keratinocyte migration and an enhanced dermal-epidermal junction protein deposition. It is possible to hypothesize that the effects reported by the use of IL-1β primed MSCs are due to the release of EVs containing growth factors, cytokines, and extracellular proteins [101].

In selected animal models of diseases, MSC supernatant EVs showed a similar therapeutic effect of MSCs alone, confirming their role in tissue repair [89,102]. 

## 5. Extracellular Vesicles (EVs)

It is now well demonstrated that cell-to-cell communication is fundamental for multicellular organisms both during development and also in adult tissues. This communication can be mediated either by direct cell-to-cell contact, realized by cell adhesion molecules (CAMs), or by transfer of secreted molecules, kept inside EVs [103]. For many years since their discovery [104,105] EVs have been considered as cellular debris and their study was overlooked until the 21st century due to insufficient research techniques. Today, it is now accepted that almost all cell types, included stem cells, release EVs through an active and specific pathway, and they have been found in almost all the body fluid (e.g., serum, plasma, milk, cerebrospinal fluid, saliva and urine) [106,107,108].

EVs consist of a heterogeneous population of vesicles surrounded by a lipid bilayer containing transmembrane proteins, DNA, RNA, and other metabolites. The lipid envelope protects bioactive molecules from extracellular enzymatic degradation. EVs were classified into exosomes, microvesicles and apoptotic bodies according to their biogenesis, surface markers, function and size (Figure 1, Table 1) [109,110]. Exosomes are derived from the multivesicular endosomal system and are secreted when the multivesicular bodies are fused with the plasma membrane. Their size is below of 50 nm in diameter and they are enriched in endosome-derived components (e.g., Alix, Tsg101) [109,111]. On the contrary, microvesicles (MVs) arise directly from the outward budding of the plasma membrane, followed by a fission event similar to the abscission step observed in cytokinesis [112]. These particles range in size between 50 nm and 1 μm [109]. MVs formation requires increased cytosolic calcium ions, molecular rearrangements within the plasma membrane in terms of lipid and protein composition (e.g., phosphatydylserine exposure), and degradation of membrane cytoskeleton [113,114,115]. Conversely, apoptotic bodies are larger than exosomes and MVs [116], and are produced by the blebbing of aging or dying cells [117,118]. However, since this last type of EVs is less studied, they will not be discussed further. 

Recently, as some confusion existed in the literature regarding the terms used by the researchers to describe vesicles present in the extracellular space (e.g., exosomes, shedding vesicles, melanosomes, prostasomes, ectosomes, microvesicles, argosomes, oncosomes), the International Society for Extracellular Vesicles (ISEV) have recommended researchers to use the term “extracellular vesicles (EVs)” for all types of vesicles, as the generic term for particles naturally released from the cell that are delimited by a lipid bilayer and cannot replicate, i.e., do not contain a functional nucleus [119]. Following this recommendation, we use the term EVs throughout the entire paper.

Today EVs are considered key elements of cell-cell communication, containing not only bioactive molecules, but also whole organelles (i.e., mitochondria and ribosomes) [120,121]. 

## 6. Extracellular Vesicles Composition 

The composition of EVs depend mainly on their cell origin and on the stimuli they receive, under both physiological and pathological conditions. Indeed, it is well recognized that targeting of bioactive molecules within EVs is a selective process, being specific molecules included or excluded [122,123,124,125]. After their release into the extracellular environment many EVs rapidly break down, liberating their cargo in the surrounding microenvironment. Otherwise, EVs can deliver their cargo to target cells via different pathways: direct interaction, resulting in EV fusion with the target cell membrane or in their endocytosis, and interaction mediated by ligand-receptor binding [126,127,128] (Figure 2). 

Multi-omics studies allowed to characterize EVs content, essential for their biological functions. Proteomic analysis of EVs have demonstrated that proteins released through vesicles are involved in intercellular communication and play a pivotal role in cell signaling, differentiation, cell adhesion, angiogenesis, and apoptosis. They include growth factors, cytokines, chemokines, extracellular matrix proteins, metalloproteinases, etc. [129]. In particular, it has been demonstrated that cells use EVs for unconventional protein export. In fact, some cytosolic proteins (e.g., FGF-2, IL-1β) which lack exocytosis signals necessary for the classical endoplasmic reticulum-Golgi secretory pathway are released within EVs [130,131]. 

One of the first proteomic characterization of MSCs was performed in 2012 by Lai et al., and Kim et al. [132,133]. These authors by two different methods identified 857 and 730 proteins, respectively. Among the common subset of 315 proteins, both cytoplasmic and membrane proteins were identified. Specific markers of MSCs, such as CD63, CD9, CD109, CD151, CD81, CD248, and CD276, as well as surface receptors (e.g., PDGF-RB, EGF-R), signaling molecules (RHO, CDC42, MAPK1, Wnt5B) responsible for self-renewal and cell differentiation, have been also identified [133]. In addition, proteins implicated in EV biogenesis, intracellular trafficking and fusion were characterized (RAB1A, RAB2A, RAB5A/B/C, RAB7A, RAB8). Finally, Kim et al. identified proteins implicated in cell adhesion, migration, and morphogenesis [133]. 

Nowadays, there are three public databases containing data from proteins, mRNAs, miRNAs, and lipids enclosed in prokaryotic, non-mammalian eukaryotic, and mammalian EVs: EVpedia, Exocarta, and Vesiclepedia [134,135,136].

One of the best characterized functions of MSC-derived EVs is to improve angiogenesis, whose efficiency varies according to their tissue of origin. The enriched angiogenic proteins identified in EVs include VEGF, vWF, TGF-β1, IGF-1, and IL-8 [137,138,139]. Literature data have demonstrated that conditions such as hypoxia, serum deprivation, exposure to IFN-γ, TNF-α, are able to influence EV angiogenic factors concentration. For example, serum deprivation and hypoxia increase the release of VEGF, FGF-2, HGF, IGF, and TGF-β within EVs derived from ADSCs, bone marrow MSCs and placenta MSCs [95,99].

Recent attention has also focused on the capacity of EVs to induce epigenetic changes in target cells [140,141,142,143]. These changes could be due to the transfer through EVs of genetic information between donor and target cells, making them similar to the producing cells. Indeed, it is now well demonstrated that EVs in addition to growth factor and other proteins also contain nucleic acid, especially RNA (i.e., mRNA, miRNA, tRNA, lncRNA). 

Particularly important is the transfer of miRNA, which are small noncoding RNA that are 19 to 23 nucleotides long. miRNAs are powerful regulators of gene expression due to translational inhibition or to degradation of target mRNAs [144,145]. Detailed analysis of miRNA content in MSC-EVs demonstrated that there are profound differences between vesicles and cellular content, suggesting an active process of sorting and packaging of miRNA into EVs, as already demonstrated for protein sorting [146]. Several of the identified miRNAs are involved in angiogenesis modulation. They target and modulate the expression of regulatory angiogenic genes encoding for MMPs and growth factors such as VEGF, PDGF, FGF, and EGF [147,148].

Furthermore, EVs are also enriched in tRNAs [149], which contribute to maintaining stem cell potency, stimulating cell survival and inhibiting cell differentiation [150]. Several evidences have also demonstrated the presence of long noncoding RNAs (lncRNAs) within EVs [151]. These RNAs are able to induce epigenetic modification by binding to specific genomic loci and recruiting epigenetic regulators. In this way, EVs may induce epigenetic modifications in target cells.

EVs also contain bioactive lipids, as well as lipid metabolic enzymes. Lipids have been implicated in multiple aspects of EV biogenesis and function. 

As for proteins and RNAs, EV lipid composition is distinct from that of cells of origin. In particular, they contain cholesterol, sphingomyelin, glycosphingolipid, ganglioside GM3, and phosphatidylserine [152]. Cholesterols, sphingomyelins and phosphatidylserine are the major components of lipid rafts, with all three lipids showing increased abundance in EVs when compared to their secreting cells. The presence of lipid raft-associated proteins, including flotillin-1, suggested that lipid rafts could influence selective protein sorting into EVs. They also transport arachidonic acid, phosphatidic acid, prostaglandins, leukotriens [153], and are enriched in spingosine-1-phosphate, a signaling lipid that with the other lipids of the EVs induce several biological responses, including cell proliferation and migration [154]. Lipid composition is also fundamental to determine EV-target cell interaction [155] and can be influenced by MSC culture conditions. 

## 7. Extracellular Vesicles-Based Regenerative Medicine

As described above, it appears evident that positive effects of MSC transplantation is most likely due to their paracrine activity, rather than to their homing and differentiation in the injured tissues. According to these observations it is conceivable that EVs may replace the stem cells used in regenerative medicine, playing their biological functions. 

Recently, more and more in vitro and in vivo studies have demonstrated that stem cell conditioned media alone, and in particular EVs, can replicate the reparative effects observed by using stem cells (Figure 3) [156,157,158].

Most of the literature concerning EVs released by MSCs is isolated from different tissues. That is why we barely mention the topic and for more in depth information we refer to some recent reviews [159,160]. 

Many studies showed that the protective effects exerted by MSC-EVs in the respiratory injury model are dependent on the regulation through miRNAs of inflammation and proliferation [161,162,163]. It is interesting to know that in a mouse model of hemorrhagic shock and laparotomy-induced lung injury, Potter et al. demonstrated that both MSCs and EVs ameliorate the lung condition, with several differences in their molecular effect [164]. 

EVs have been proved to be a good candidate to substitute MSCs in mild acute and acute kidney injury, in particular by inhibiting epithelial cell apoptosis, promoting wound regeneration, and by stimulating angiogenesis [165,166,167,168,169]. MSC-EVs are also efficient in chronic kidney diseases [170,171,172,173]. It was also demonstrated that, in acute renal injury, EVs both in vivo and in vitro are responsible for inflammation reduction through the expression of the receptor CCR2, which reduces migration and activation of macrophages [174]. Autophagy induction is another mechanism used by MSC EVs to ameliorate tubular function [175,176].

The protective effects of MSC-EVs in cardiovascular diseases have been already observed in 2010 [177]. It is interesting to note that in some cases, the effects of the EVs were significantly better than those of MSCs [178]. Generally, EVs can promote myocardial regeneration and restore cardiac function by directly activating cardiac precursor cells and inducing their differentiation, through the promotion of cardiomyocytes survival and proliferation. EVs are also able to inhibit cardiomyocytes apoptosis, to stimulate angiogenesis, and finally to regulate the inflammatory response in the damaged area [179,180,181,182]. One of the possible mechanism is related to the increase in cardiomyocytes of Bcl-2 expression, responsible for reduction of cardiac cell apoptosis, promotion of proliferation and angiogenesis [183]. Apoptosis reduction could also be due to autophagy activation via the PI3K/Akt/mTOR pathway [184]. Similarly increased angiogenesis is induced by the presence within EVs of the extracellular matrix metalloproteinase inducer EMMPRIN, VEGF, MMP9 and miRNA-210 [185,186]. Other miRNAs involved in myocardium protection are miRNA-21a-5p, miRNA-21, miRNA-125-5p, and miRNA93-5p [187,188,189,190]. Besides, overexpression of miRNA-30d inhibited autophagy and promoted polarization of macrophages into the anti-inflammatory M2 type [182]. The expression of inflammatory factors was also down-regulated by the miRNA-126 [191].

Particularly interesting and promising are the results obtained by using MSC-EVs in neurological and neurodegenerative diseases. In fact, differently from MSCs, EVs because of their lipid structure can cross the blood-brain barrier [192,193]. In addition, they have an extremely low immunogenicity that allow them to exert an immunoregulatory role avoiding clearance to which MSCs are subjected. Doeppner et al. showed that MSC-EVs in stroke treatment had the same tissue regeneration capability as MSCs [193]. In an another study, EVs had even better recovery effects in stroke damage repair, most likely due to their higher blood-brain barrier permeability than MSCs [194]. EVs have been also successfully used in brain trauma. Indeed, Chuang et al. discovered that MSC-EVs could promote nerve and endothelial cell regeneration, increase VEGF production involved in angiogenesis [195], and also reduce the local inflammatory response [196]. A reduction of the inflammatory response was observed by Kim et al. too [197,198]. A diminished inflammation was even observed in spinal cord injury, most likely via neuronal apoptosis inhibition, angiogenesis stimulation, A1 astrocytes inactivation and up-regulation of anti-inflammatory IL-10 and TGF-β as well as down-regulation of the pro-inflammatory TNF-α and IL-1β [184,199,200,201,202]. An essential function after spinal cord injury and brain ischemia was also attributed to the miRNA-113b [197,203].

## 8. Induced Pluripotent Stem Cell Derived EVs in Regenerative Medicine

Despite all the advantages so far considered for the MSCs, and although MSCs are used to treat several kinds of diseases completely different from each other, they remain multipotent stem cells, and their ability to differentiate in some cell types (e.g., neural precursor cells) is reduced. Furthermore, the number of MSCs that can be obtained by a single donor is limited. On the other end, the use of pluripotent embryonic stem cells carries potential ethical problems with respect to their clinical use. For this reason, the development of iPSCs opened a new scenario in regenerative medicine. Despite their potentiality, clinical application of iPSCs is limited because of their tumorigenic potential, low engraftment, and cell retention. 

As described above, more and more in vitro and preclinical studies have demonstrated that stem cells exert their regenerative potential via paracrine mechanisms, which include EV release. Due to their lower immunogenicity, easier handling, and having no risk of tumor formation, researchers recently focused on their potential as possible alternatives to stem cells [159,204,205,206], studying experimental model diseases treated with MSC-EVs. To date, a total of eight clinical trials on the application of EVs for disease treatment were identified [160]. Accordingly, many researchers investigated the role of EVs released by iPSCs in tissue regeneration. Several proteomic analyses have identified cytokines, chaperones, signaling molecules, plasma membrane proteins, isolated from human pluripotent stem cells [207,208], and it is reasonable to hypothesize that iPSC-EVs contain many kinds of these proteins. Therefore, many researchers shifted their attention to the EVs released by transplanted cells. To date, many studies have highlighted the importance of EVs and their miRNA in cell-to-cell communication within the cardiovascular system, also because the myocardial infarction is a leading cause of death worldwide [209,210]. Wang et al. demonstrated that iPSC-EVs can be taken up by cardiomyocytes in vitro, producing cytoprotection against H_2_O_2_ induced oxidative stress. Additionally, in vivo EV transplantation protected cardiac cells by inhibiting their apoptosis via caspase 3/7 inhibition [211]. These data raise the possibility that iPSC-EVs could be protective for cardiomyocytes. EV cytoprotective effect is also related to their miRNA content, in particular miRNA-21 and miRNA-210. Xuan et al. generated cardiac progenitor cells (CPCs) starting from iPSCs by treating them with the small molecule ISX-9, with antioxidant and regenerative properties, and used EVs derived from these cells to repair post-infarcted heart [212]. They found that the cardioprotective EV effect was in part due to their miRNA content. In particular, CPC^ISX-9^-derived EVs were highly enriched with miRNA-520/-373 family members. miRNA-373 was originally identified as a human embryonic stem cell specific miRNA and is involved in the regulation of cell proliferation, apoptosis, senescence, migration and invasion following hypoxia [213]. miRNA-373 inhibited TGF-β, hypoxia-induced fibrotic genes upregulation (i.e., GDF-11 and ROCK-2), and myofibroblast differentiation in fibroblasts. Additionally, in vivo data confirmed the therapeutic role of CPC^ISX-9^-derived EVs. EVs were responsible for cardiomyocyte proliferation and angiogenesis, too. These studies confirmed that EVs can be used for the treatment of ischemic heart by overcoming the limitation of cell based therapy [213].

Oh et al. demonstrated that exosomes derived from human iPSCs exert a protective role on skin fibroblasts against photoaging and natural senescence [214]. They found that iPSC-EVs stimulated both proliferation and migration of dermal fibroblasts, increased the expression level of collagen type-I in photoaged and senescent fibroblasts, reduced the expression level of MMP-1/3 during senescence [214]. 

As described above, several data have demonstrated the beneficial effects of MSC-EVs in both acute and chronic kidney disease. Lee et al. have already demonstrated that also iPSCs can protect the kidney through the induction of an anti-apoptotic, anti-inflammatory, and antioxidant response [215]. Collino et al. evaluated the protective and regenerative potential of iPSC-EVs in an acute kidney injury model to explore the possibility of their use in kidney regenerative medicine [216]. In their study, the authors demonstrated the main mechanism associated with mitochondria protection. Indeed, after EV administration, the mitochondrial mass of renal cells was maintained as well as the ΔΨm, fundamental ATP generation, and cell survival. Even if this protective mechanism is similar to that exerted by ADSC-EVs, IPSCs-EVs showed a higher efficiency in renal cell protection. Furthermore, iPSC-EVs have been involved in immune response modulation, with a decrease in infiltrating macrophages, and in inflammation modulation, through macrophage M2 polarization. A modulation of oxidative stress was also observed [216].

Furthermore, iPSC secretome were used to treat pulmonary fibrosis. In particular, Gazdhar et al. have demonstrated that the secretome induces alveolar epithelial repair in vitro and also reduces lung fibrosis in vivo [217]. 

## 9. EVs as a Cargo Delivery System

Several data have demonstrated that EVs can be engineered for nucleic acid delivery to target cells, using either cell preconditioning or exogenous transfection, especially resulting in an enrichment of natural miRNAs.

Next generation sequencing revealed that EVs originating from hypoxic human neural stem cells have an increased levels of 53 miRNAs and a reduction in 26 miRNAs [218]. Similarly, microarray analysis of EV released by cardiac progenitor cells under hypoxic stress condition showed an upregulation of 11 miRNAs compared to EVs secreted from normoxic cells [219]. Moreover, Xiao et al. have demonstrated that EVs enriched in miRNA-21, released by cardiac progenitor cells stress-treated via H_2_O_2_ incubation, protected cardiomyocytes from damage in ischemic conditions [220]. 

However, preconditioning offers only a limited control in the specific types of enriched miRNA, and standardization also remains a problem, as different conditions can lead to different types and levels of cargo enrichment. 

Conversely, transfection or transduction, an indirect method of cargo loading, can offer more much control than preconditioning. Han et al. have transfected umbilical cord MSCs with miRNA-675 mimic, leading to the secretion of EVs that showed an increased expression of this miRNA [38]. When they injected the miR-675-enriched EVs delivered in silk fibroin hydrogels in a murine model for aging-induced vascular dysfunction, they observed a promotion of blood perfusion in ischemic hindlimbs and reduction of proinflammatory molecules expression [221]. Furthermore, lentiviral transduction have used to engineer MSCs to overexpress, for example, miRNA-let7c, and secrete EVs enriched in this miRNA [186]. The enriched EVs were showed to mediate the improved transfer of miR-let7c to the kidneys, attenuating renal fibrosis.

In addition to miRNA, siRNAs are another class of small RNAs which have been exogenously loaded within EVs. siRNA loading is similar to that of miRNA loading, with electroporation being the most common method of active loading [222]. In a model of acute lung injury, EVs isolated from iPSCs derived by reprogramming renal epithelial cells were subjected to electroporation to introduce siRNA against ICAM-1, which is involved in the inflammatory response [223].

## 10. EVs Functionalized Scaffolds in Regenerative Medicine

In recent years, regenerative medicine has gone a step further thanks to the use of engineered constructs (i.e., scaffolds) [224]. Several biomedical researches have focused on the use of scaffolds functionalized with EVs, trying to improve regenerative capacity. In particular, EVs used to engineer different types of scaffolds are those derived from MSCs due to their proangiogenic properties.

EVs can be immobilized on a variety of polymer-based scaffolds. For example, Xie and co-workers coated decalcified bone matrix (DBM) with bone marrow-derived MSC EVs using fibronectin as the immobilizing agent. The authors showed that the implanted EV-functionalized scaffolds promoted bone regeneration in a murine model. Significant new bone matrix was observed, with higher levels of CD31-positive vessels within the matrix compared to unmodified scaffolds, confirming the ability of the EVs to promote bone formation and vascularization [225].

Similar results were obtained with EVs derived from other stem cell sources combined with other type of scaffolds. Li and collaborators have used a PLGA/pDA (poly lactic-co-glycolic acid) scaffold engineered with hADSC-EVs in the murine model of critical-sized calvarial bone defects. They observed a significantly higher bone tissue and mature collagen formation in hADSC-EV modified scaffolds compared to the unmodified ones [226].

To stimulate wound healing, EV-modified sponges and patches were used. Shi et al. successfully loaded gingival MSCs, onto a chitosan/silk hydrogel sponge, to cover surgically-induced skin in diabetic rat [227]. Quantification of wound size showed that the EV-modified sponges had a significantly greater wound closing ability because of their aptitude in angiogenesis and collagen deposition [227]. 

EV-modified hydrogel patches begun to emerge as a therapeutic strategy for cardiac recovery and regeneration. EVs secreted from iPSC-derived cardiomyocytes were encapsulated in hydrogel patches. Their application to rat myocardium promoted arrhythmic recovery, decreased cardiomyocyte apoptosis after infarction, and reduced infarct size and cell hypertrophy [228]. 

## 11. Vantages and Disadvantages in the Use of Stem Cell-EV Based Therapies

To date, more and more data have demonstrated that EVs can substitute stem cells in regenerative medicine and tissue regeneration. EVs display several advantages over cells; (1) they do not solicit host immune response due to their low immunogenicity, so there is no need to match the donor and the recipient. (2) they can be constantly released by immortalized stem cells, overcoming the low amount of cells isolated, especially adult ones, compared to the large number needed for therapy; (3) they avoid tumor formation because of their inability to self-replicate; (4) due to their lipidic structure EVs are easily stored for a long time at −80 °C without losing their bioactivity, in addition to being more resistant to freezing and thawing than stem cells; (5) EVs can cross some barriers, such as blood brain barrier [229], and, because of their dimensions, move into wound areas and capillary; (6) there is no risk of infection transmission; and, (7) EVs containing specific cargo (i.e., nucleic acids, proteins) can easily obtained. Indeed, it is possible to manipulate EV releasing stem cells to improve their beneficial effect. EV content can be modified for example by hypoxia and stress preconditioning [219], serum deprivation, or by genetic modification and epigenetic reprogramming of EV-producing stem cells. In addition, genetically engineered EVs containing small therapeutic molecules can be used as a delivery system [222]. Exposure to TNF-α or IFN-γ also modifies EV content. 

Even if stem cell isolated EVs are much safer than cell therapy for bench-to-bedside translation, there are also problems with their use; (1) contrasting data regard the duration of EV effects, as it is not clear regarding their half-life. On the contrary, engrafted donor stem cells can continuously release EVs by having long term beneficial effect; (2) in some cases (e.g., iPSC-EVs) EVs stimulate fibroblast proliferation increasing scar tissue formation; (3) possible inefficient endocytosis by target cells has to be evaluated; (4) EV properties can vary between cell batches and passage number, making standardized therapies difficult to manufacture; (5) the current standard methodologies for EVs isolating involves long culture times to obtain hundreds of millions of cells and intensive ultracentrifugation steps, all of which are expensive and require a great deal of time.

Before using EVs for a cell free therapy, some aspects must be considered. It is necessary to define univocal methods to select the most appropriate source for their release. In fact, different cell sources release EVs with different compositions, which affect the effects on different diseases. Only a few studies have identified the whole proteome contained in MSC-EVs, and the comparison clearly indicates several differences in their composition [132,133]. It is also essential to develop a unified protocol of cell culture conditions, quantification, and for collecting EVs. In particular, regarding EV isolation, the techniques most commonly used are ultracentrifugation and polymeric precipitation, but they permit the coprecipitation of EVs and culture medium components; in addition, the recovery is very low. To be used in clinical trials, EVs must have specific requirement, such as purity and sterility above all, but also their composition and potency need to be determined [230]. Biodistribution of EVs need to be investigated. 

Additionally, due to the still limited number of clinical trials using EVs for tissue regeneration the safety and the optimal dose should be tested and GMP procedures have to be set up.

## 12. Conclusions and Future Perspective in EV-Based Therapies in Regenerative Medicine

The data summarized in this review suggest that EVs could be used in regenerative medicine in a cell free therapy, as they mimic stem cell effect. 

Their potential in tissue regeneration is mainly due to their ability to regulate the host immune response, promote the migration and the proliferation of damaged cells, tissue neovascularization. Preliminary EVs positive effect were already observed in skin, heart, kidney, lung reconstruction, and also in neurodegenerative diseases. 

Even if EVs-based tissue engineering represents a promising clinical strategy in the field of regenerative medicine, and many studies on EVs have improved our knowledge on their origin, there are still many challenges to overcome. The researchers proposed distinct methods for EV isolation, although different culture conditions, such as FBS, supplements, cell seeding density, age of the donor cells, and oxygen concentrations, may affect EV production and composition. Therefore, before their use in clinical trial, a standard protocol needs to be formulated for EV recovery. Furthermore, to improve the therapeutic effects of EVs it is necessary to improve the methods to edit stem cells to introduce within them specific molecules (i.e., miRNAs and lncRNAs). 

## Figures and Tables

**Figure 1 cells-09-01163-f001:**
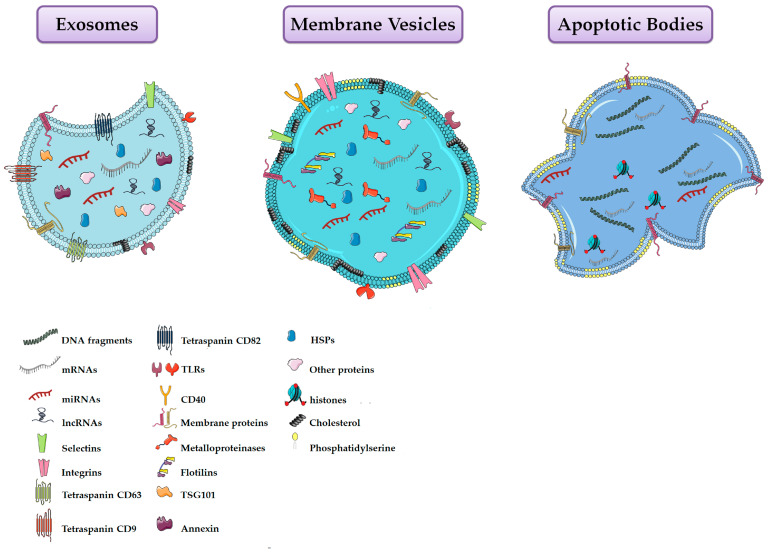
Schematic representation of the three main types of extracellular vesicles released by cells.

**Figure 2 cells-09-01163-f002:**
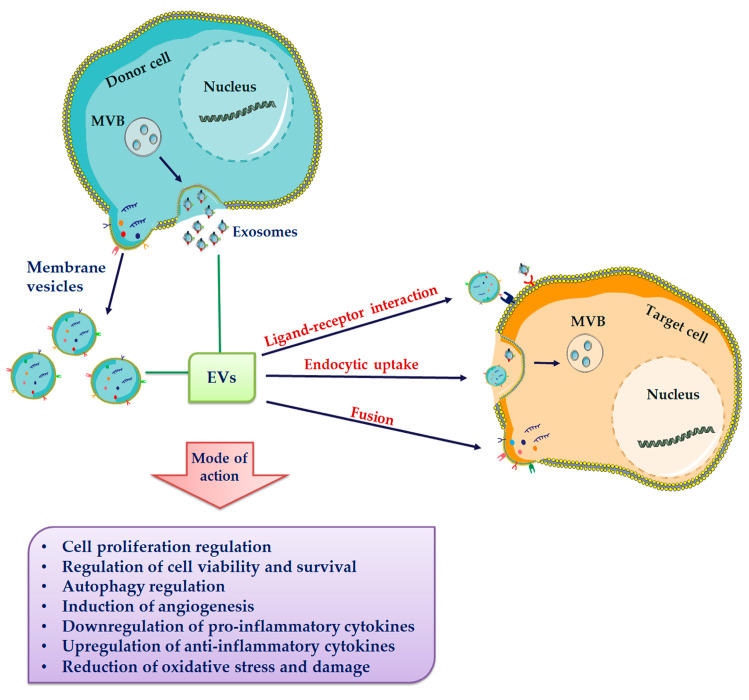
Extracellular vesicle (EVs) release and uptake by target cells. EVs, including exosomes and membrane vesicles, are released in the extracellular milieu. All subtypes of EVs share a general composition with an outer lipid bilayer and various proteins, lipids, and nucleic acids. EVs have been suggested to be internalized into target cells by various uptake mechanisms including membrane fusion, different endocytic pathways and receptor-mediated endocytosis.

**Figure 3 cells-09-01163-f003:**
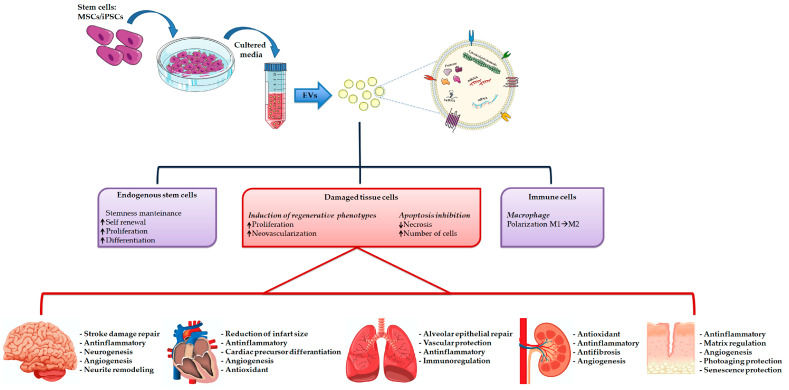
Extracellular vesicles isolation and their therapeutically potential in tissue repair.

**Table 1 cells-09-01163-t001:** Summary of extracellular vesicles properties.

Property	Exosomes	Membrane Vesicles	Apoptotic Bodies
Size	30–120 nm	100–1000 nm	50–4000 nm
Morphology	Homogenous cup-shape	Heterogeneous irregular	Heterogeneous irregular
Buoyant density	1.13–1.19 g/cm^3^	Not well defined	1.16–1.28 g/cm^3^
Origin	Endosomal	Plasma membrane	Apoptotic cells
Proteins	CD63, CD81, CD9, annexins, heat-shock proteins, Alix, Tsg101, clathrin, caveolins, integrins, TfRs	Integrins, flotillins, selectins, CD40, metalloproteinases	Histones
Lipids	Lysobisphosphatidic acid, cholesterol, ceramide, sphingomyelin and low concentration of phosphatidylserine	High amount of cholesterol, sphingomyelin, ceramide, high concentration of phosphatidylserine	High concentration of phosphatidylserine
Nucleic Acids	mRNA, lncRNAs and miRNAs	mRNA, lncRNAs and miRNAs	mRNA, miRNA, fragments of DNA

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
