# Peer review of "Mesenchymal and Induced Pluripotent Stem Cells-Derived Extracellular Vesicles: The New Frontier for Regenerative Medicine?"

_cells, 2020, doi:10.3390/cells9051163_

Round 1

Reviewer 1 Report

In the manuscript titled “Extracellular vesicles: the new frontier of stem cell regenerative medicine?”, authors summarized the current applications of extracellular vesicles (EVs), especially focusing on the stem cell-derived EVs in regenerative medicine. Generally speaking, the review covers the most updated investigations and the manuscript is well-written in a logical way. The authors also included three figures and one table, which make the content easier to understand.

One concern is that the International Society for Extracellular Vesicles consensus recommendation on nomenclature is to use EV as the generic term for particles naturally released from the cell. However, Figure 1 and Table 1 still used the previous nomenclature. The authors need to clarify this in the manuscript.

Reviewer 2 Report

The present manuscript reviews the state-of-art on the therapeutic potential of extracellular vesicles (EVs) that are released by some stem cell types, in particular in the contexts of regenerative applications. The main focus of this review is mesenchymal stem cells (MSCs) and their paracrine effects via extracellular vesicles, and induced pluripotent stem cells (iPSC) as an alternative source of EVs. This manuscript provides an in-depth inventory of current knowledge and is supported by a very complete bibliography, which provides an interesting contribution to understanding the field.

The aspects details below may improve the manuscript soundness and positioning within the general domain of stem cell-based tissue regeneration:

1) Considering the focus of the review that is essentially on EVs from MSCs and iPSCs, abstract (and eventually title) should be revised to mention these topics clearly.

2) The second and third paragraphs (Stem cells / Stem cells and regenerative medicine) provide general overview of these domains. In these fields, epidermal / keratinocyte stem cells constitute a pioneer system, as they have proved efficacy in the bioengineering of three-dimensional skin substitutes (regenerated organoids) since more than three decades, with major clinical achievements, notably graft for skin replacement in burn patients, and more recently regeneration genetic correction of the entire epidermis in a patient suffering from a genodermatosis.

A paragraph should be added on epidermal / keratinocyte stem cell-based regeneration, summarizing the key clinical applications, and actual researches conducted to move the domain forward. I would suggest making reference to the works listed above

Skin substitute graft in burn patients:

- Gallico et al., N Engl J Med. 1984 Aug 16;311(7):448-51.

- Ronfard et al., Transplantation. 2000 Dec 15;70(11):1588-98.

Use of keratinocyte stem cells for cell and gene therapy (epidermolysis bullosa patient):

- Hirsch et al., Nature. 2017. doi: 10.1038/nature24487.

Search for ‘stemness’ targets to improve keratinocyte stem cell-based grafts bioengineering:

- Fortunel et al., Nat Biomed Eng. 2019. doi: 10.1038/s41551-019-0464-6

3) A link between MSC uses in regenerative medicines, and the domain skin regeneration by ex vivo bioengineered epidermis substitute grafting. This proposed approach should be mentioned.

MSC use improves epidermis substitute engraftment:

Magne et al., J Invest Dermatol. 2020 doi: 10.1016/j.jid.2019.07.721

4) A major clinical application of MSC use is the attenuation of the tissue damages induced in severe radiation burns, in which paracrine effects are believed to play a role, and for which the possible use of extracellular vesicles is seriously considered. A paragraph should be included to review this domain and make reference to the works listed above.

Human patients:

- Lataillade et al., Regen Med. 2007. Sep;2(5):785-94.

- Bey et al., Wound Repair Regen. 2010. doi: 10.1111/j.1524-475X.2009.00562.x

Pig models:

- Linard et al., Stem Cell Res Ther. 2018. doi: 10.1186/s13287-018-1051-6

- Linard et al., Stem Cells Transl Med. 2013. doi: 10.5966/sctm.2013-0030
